# Sim2real transfer learning for 3D human pose estimation: motion to the rescue

**Carl Doersch**[*]      **Andrew Zisserman**[*†]

[*] Deepmind, London      [†] VGG, Department of Engineering Science, University of Oxford

## Abstract

Synthetic visual data can provide practically infinite diversity and rich labels, while avoiding ethical issues with privacy and bias. However, for many tasks, current models trained on synthetic data generalize poorly to real data. The task of 3D human pose estimation is a particularly interesting example of this sim2real problem, because learning-based approaches perform reasonably well given real training data, yet labeled 3D poses are extremely difficult to obtain in the wild, limiting scalability. In this paper, we show that standard neural-network approaches, which perform poorly when trained on synthetic RGB images, can perform well when the data is pre-processed to extract cues about the person's motion, notably as optical flow and the motion of 2D keypoints. Therefore, our results suggest that motion can be a simple way to bridge a sim2real gap when video is available. We evaluate on the 3D Poses in the Wild dataset, the most challenging modern benchmark for 3D pose estimation, where we show full 3D mesh recovery that is on par with state-of-the-art methods trained on real 3D sequences, despite training only on synthetic humans from the SURREAL dataset.

## 1  Introduction

3D pose estimation, especially for humans, is a classic computer vision problem, with applications in imitation learning, robotic interaction, and activity understanding. Pose estimation is extremely challenging with objects that are articulated, deformable, or have wide intra-class variation, as is the case with humans. Therefore, state-of-the-art approaches rely on neural networks and learning. However, learning-based methods are extremely data hungry, and acquiring sufficient data in the real world is difficult. First, there is no straightforward way for people to annotate 3D ground truth poses. Worse, in settings like industrial warehouses or homes, hundreds of thousands of different object types may appear, and new objects may arrive at random. Here, even simple labeling will generally be impractical, much less 3D poses. And any time data involves real humans, issues with privacy, intellectual property, and bias can become serious obstacles [33, 36, 92].

Synthetic data, however, provides an answer to all these problems, providing a potentially infinite dataset where ground-truth properties are easily accessible. In domains with many objects where labeling is impractical, scanning and simulating objects may not be [25, 26]. Furthermore, synthetic humans do not have any privacy or intellectual property concerns [33], and datasets can be balanced exactly with respect to sensitive attributes like race, gender, and other physical characteristics, minimizing the problems algorithms currently have with bias [36, 92]. Even better, simulations can be made interactive for training robotic policies.

Considering all the advantages, why isn't simulation the dominant approach in computer vision? One problem is that neural networks trained on synthetic data do not necessarily work on real data as well as methods trained directly on real data. Thus, even though such "sim2real" transfer has performed well in some domains, such as hand tracking [48] or text detection [23], it is rare on the most popular benchmarks like human pose estimation, object classification, or object detection. Curiously, the

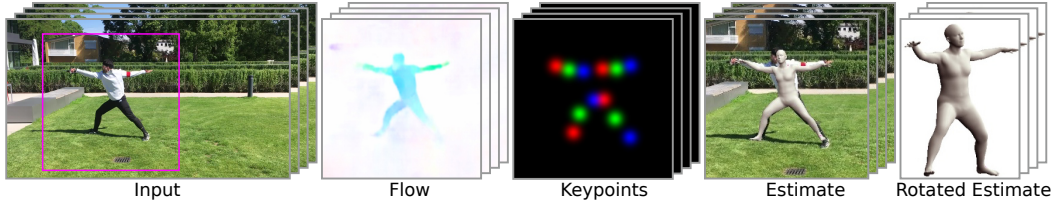

|  Input | Flow | Keypoints | Estimate | Rotated Estimate |

Figure 1: Given a detected person in a video sequence, our algorithm estimates 3D pose by first extracting optical flow and keypoint location estimates. We find that neural networks trained on such input can generalize well even when trained purely on synthetic inputs of this type, far better than equivalent networks trained on only RGB synthetic images.

community's gold standard for popular vision algorithms is generally *quantitative performance on large evaluation datasets*. In practice, this kind of evaluation is limited to tasks where labels are easy to obtain, and for such tasks, it is equally straightforward to create large training sets with matching statistics. In contrast, vision problems like robotic manipulation—where simulation *has* been influential—are less popular not because they are unimportant, but because they exist in a sort of "evaluation blind spot," due to the lack of standard benchmarks that can boil real-world performance down to a number. 3D human pose estimation is a rare exception to this trend: manual annotation is almost impossible, yet a small dataset of in-the-wild data is available for evaluation due to clever use of external sensors [87], called 3D Poses in the Wild (3DPW). Thus, we set our sights specifically on this problem, as a testbed for understanding how to design algorithms that can learn real-world pose estimation from simulation.

To train a 3D human pose estimation network, we must first confront the domain gap posed by standard datasets. The synthetic humans video dataset SURREAL [86], for example, lacks deformable clothing meshes, realistic lighting, and environmental interaction. While these aspects could be improved via better simulators, the problems with SURREAL are representative of the problems that rapidly-scanned real-world objects have: small features may be lost, and physical properties will be approximate. Humans have little difficulty understanding the 3D structure of SURREAL without any prior experience, giving hope that computers can transfer between the domains as well. However, we find empirically that naïve transfer for computers is poor from SURREAL to 3DPW.

Our key insight is that *motion*, extracted from video sequences, can be a better cue for enabling transfer. Our intuition is grounded in the psychology literature, where humans have been shown to extract remarkably detailed 3D interpretations when only simple point-light motion is visible [13, 32, 39]. Modern simulations (including SURREAL) explicitly use 3D models which match the 3D geometry of real humans, and therefore hypothetically match well in terms of plausible motion.

Armed with this intuition, we build a system to estimate 3D human poses in real videos. Our core contributions are relatively simple modifications to a standard 3D human pose estimation algorithm— Human Mesh Recovery (HMR) [34]—which greatly improve transfer from simulation to reality. Specifically, we first modify SURREAL to contain more realistic overall motion, for example, by compositing SURREAL humans onto real backgrounds from videos collected in-the-wild. Then we add explicit motion cues, including optical flow from FlowNet [15] (also trained with synthetic data), and 2D keypoint tracks obtained from an off-the-shelf 2D detector (such as [53], which is trained on real 2D keypoints, and therefore used only at test time, while at train time the 2D keypoints come from the simulator). We find that both modifications substantially improve performance on 3DPW, tracking close to state-of-the-art performance, while adding synthetic RGB inputs can actually harm performance. We also compare to the standard Domain Adversarial Neural Network approach to domain transfer [18], and find relatively marginal benefits compared to motion cues.

## 2   Related Work

Our work is part of a long line of research that has attempted to use simulation for human 3D pose estimation. Principal among these is work on using datasets of synthetic humans for human pose estimation [11, 16, 20, 51, 67, 75, 84, 86, 97]. These works generally note that transfer is a challenge, and therefore the majority train on real data as well as synthetic using a variety of strategies. For

instance, some work constructs 3D datasets entirely by stitching together 2D images [67]; other works use feature selection [51] and stage-wise training [16] to improve transfer. Algorithms trained entirely on synthetic data often underperform those trained entirely on real data, even when the real datasets are small [86]. An interesting exception is work that uses depth images [72], where sim2real 3D human pose estimation is effective, although depth cameras are required.

Numerous other areas of computer vision have made use of synthetic humans with varying success. Among work on 2D pose estimation [59, 61, 69], FlowCap [69] is particularly relevant due to its reliance on flow, although its model-based optimization renders the algorithm somewhat brittle. Other works consider pedestrian detection [59–61] and action recognition [62, 63]. 3D hand pose estimation [48, 99] and eye tracking [73] are particularly promising, as neural networks trained on purely synthetic data are effective, perhaps because the lack of clothing makes appearance easier to model. Again, depth has proven useful [49, 76, 80], where state-of-the-art algorithms typically incorporate some form of generative model in-the-loop at test time.

Robotics is a particularly inspiring domain for sim2real research, and here it has been again found that more abstract representations than RGB, such as segmentations [50, 94], can improve performance. In some cases, these abstractions can be obtained automatically from a simulator alone [31]. Other works use generative models that make simulation look more like reality [9, 10], or randomize the simulator to increase the distribution overlap [70, 81]. These works emphasize that sim2real is essential: real-world data is impossible to annotate at the level of desired robot commands, and even unlabeled data is expensive since a robot can break itself or its environment. These sim2real works in robotics build on a long tradition of domain adaptation for visual data, which can involve learning maps between feature spaces [21], invariant feature extractors [18], or image-to-image translation [98]; for a review, see [12, 54].

We are also not the first to note that optical flow can be useful preprocessing for human pose estimation: optical flow has been used for 2D keypoint estimation [59, 69], part segmentation [38], and even 3D pose [4], although the latter work involves fitting a 3D model to optical flow, which is potentially slow, sensitive to initialization, and limits robustness. Similarly, some works have noted that 2D keypoints can be useful in 3D interpretation of humans [44] and objects [89]. There is also evidence that flow can aid sim2real transfer for foreground/background segmentation [82, 83].

Finally, our work is related to a long tradition of 3D human pose estimation, where learning-based methods have grown recently due to the emergence of motion-capture datasets. One straightforward approach is to 'lift' 2D poses into 3D, using either dictionaries or direct regression [3, 44, 47, 64, 79, 85, 88, 93]. Other works regress poses directly from pixels [46, 55, 56, 68, 71, 77, 78, 96], which generally relies on having a good match between training and testing. Similar to our work, state-of-the-art approaches often rely on parametric body models to incorporate strong priors on 3D poses [5, 7, 8, 22, 24, 34, 40, 43, 52, 57, 74]. Among these, we are particularly related to recent works which leverage temporal cues from videos to gain extra information about depth [6, 14, 27, 28, 35, 41, 58, 90, 91].

## 3  Algorithm

Our goal is to train a network which can predict $N$ 3D poses given a sequence of $N$ video frames. This means we first require a synthetic dataset with reasonably realistic sequences of human poses, which we obtain by compositing SURREAL renders onto real, unlabeled scenes. We also need a pose estimation model which can properly exploit motion information and, importantly, propagate this information across frames where no motion is available. For this purpose, we augment the Human Mesh Recovery (HMR) algorithm to operate on motion, and add memory in the form of a LSTM.

### 3.1  Dataset Construction

We require a dataset which captures complex human motion, provides ground-truth 3D poses for each frame, and also has all the distractors that are likely to cause problems in real data. Distractors can include background motion, occluders covering the person, and frames where the person is completely missing. While it is straightforward to get complex human motion following previous datasets such as SURREAL [86], which take pose sequences from the CMU motion capture dataset [1] and render them using the SMPL mesh [43], real videos are harder than this. SURREAL people are composited

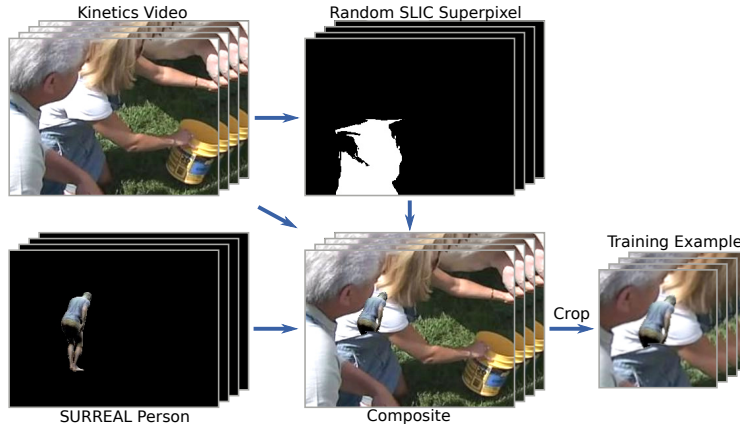

Figure 2: Procedure for generating a synthetic training example. Top left: a randomly-chosen Kinetics video. Bottom left: a SURREAL person with the background removed, to be composited onto the Kinetics video. Top right: a SLIC superpixel which was generated from the Kinetics video as a synthetic 'occluder'. Bottom middle: the composited video. Note that the person's legs are missing because they overlapped with the superpixel. Bottom right: we obtain the final training example by cropping the composited video around the person, simulating the behavior of a person detector.

on static backgrounds, which means that the video version allows for a shortcut for pose estimation: it's straightforward to segment the person from the background by identifying moving pixels. There are also no occlusions or missing frames in this dataset.

Therefore, to construct our dataset, we take humans from the SURREAL dataset and re-composite them onto a background from the large-scale Kinetics dataset [37]. Kinetics videos and SURREAL videos are sampled independently at training time: thus, $68K$ SURREAL videos may be composited on roughly $300K$ kinetics videos for around 20 billion possible combinations. Howeve, naïve compositing following SURREAL—i.e., simply removing the static background and replacing it with a kinetics video—still makes the task too easy, both because there are no occlusions or missing frames, and also because the motion of the person won't match the motion of the camera. Therefore, we modify the SURREAL video before compositing. To solve the motion discrepancy problem, our first step is to estimate the camera motion, using off-the-shelf procedures. We then translate the person to follow the camera motion, by offsetting each frame of the SURREAL video by a vector equal to the estimated camera motion of the corresponding Kinetics frame.

We then follow the procedure shown in Figure 2 to actually construct the video. Our first challenge is to simulate occlusions. One approach might be to take occluders from standard segmentation datasets like COCO [42], but this approach has some of the same problems as the SURREAL static backgrounds did. Specifically, COCO consists of static images, and so they don't have any associated motion information. We might apply random, smooth trajectories to get some motion, but occlusions in real video often have internal motion on top of global translation: i.e., the occluding objects may be deformable and have 3D structure. Furthermore, occluders in real videos tend to move with the scene. Missing either of these properties will lead to occlusions that are artificially easily identified in synthetic data, which may lead to detectors which generalize poorly on real videos. Therefore, our approach is to extract occluders *from the Kinetics video itself*. We use standard superpixel segmentation (specifically SLIC [2]) to rapidly extract segments from the video, which are generally blobs of roughly uniform color tracked throughout the video, resulting in binary masks as shown in Figure 2. One superpixel is chosen at random, and then all pixels that overlap with the SURREAL person are removed from the person, resulting in a synthetic occlusion.

As a final step, for some videos we randomly occlude the entire person for a small number of frames, which we call a 'total' occlusion. This simulates a detector failure, which is common in real-world datasets like 3DPW where the person may move out of the frame or be otherwise totally hidden. To do this, we select a continuous chunk of frames and set all feature channels to 0. We add an extra channel to the input which is 1 if the frame is totally occluded in this way, and 0 for un-occluded frames. Further implementation details on the dataset construction are given in the supplementary material.

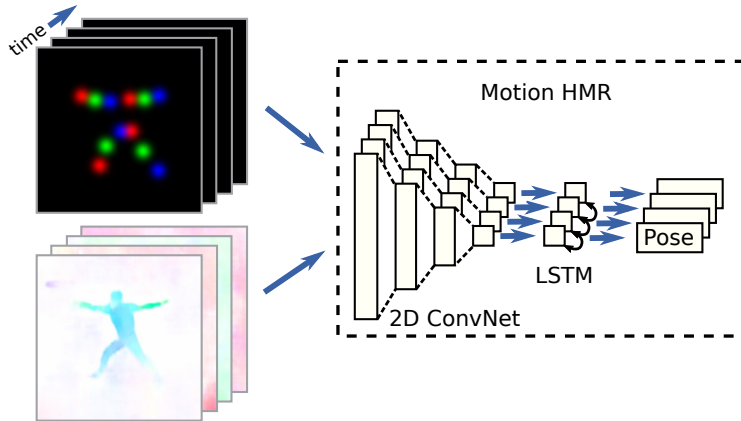

Figure 3: Network architecture for the Motion HMR model. The input is 2D keypoint heatmaps and optical flow computed on a sequence of human detections (left). These inputs are passed to a Motion HMR module (right), which applies a single-frame CNN followed by an LSTM to integrate over time, before independently estimating a pose for each frame. For baseline experiments, the inputs may be removed or replaced with RGB videos. Note, only 4 frames are shown in the schematic for clarity, but in practice 31 frames are used.

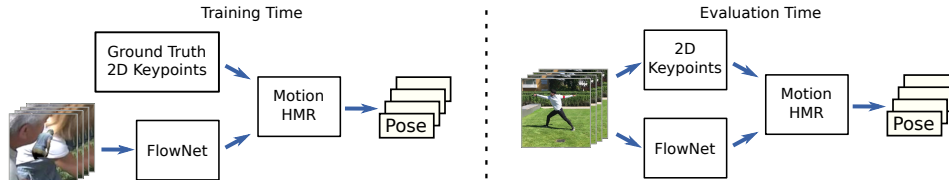

Figure 4: Our training (left) and evaluation (right) pipelines for the Motion HMR model: each takes a stack of person detections and flow from a pre-trained module (FlowNet). 2D keypoints come from simulation at train time, vs. a pretrained 2D keypoint detector during evaluation (provided by 3DPW).

## 3.2 Network Architecture

Our hypothesis is that easily-accessible motion information will be useful for bridging the sim2real gap in 3D pose estimation. Therefore, we seek a method to provide motion information to a pose estimation model, while otherwise staying close to existing pipelines for comparability. Our starting point is the Human Mesh Recovery (HMR) pipeline [34]. This recent algorithm directly regresses 3D SMPL poses from pixels, using first a ConvNet (ResNet-50) to obtain a feature vector, and then applying an iterative refinement algorithm on top of that feature vector to infer the pose.

A first modification is required to extend HMR to video. Our input, both at training and test time, is short clips (31 frames in most of our experiments). These may be the raw RGB videos, or the videos may be preprocessed to include other features like 2D keypoints or optical flow as described below. We assume that a person detector has already been run, meaning that the sequence tracks a single person whose pose needs to be estimated. A scalable memory architecture is important, because not every frame will be equally discriminative, especially when using motion features on frames that contain little motion. Thus, we need an architecture which can update its beliefs when the pose is easily identified, and otherwise leave them unchanged. We use an LSTM for this purpose (in a similar manner to [82]). Our architecture, which we call Motion HMR, is shown on the right hand side of Figure 3. This architecture applies a CNN, which is a standard ResNet-50, independently on each frame, average-pooling at the end to obtain a single feature vector per frame. We then pass these features into a bi-directional LSTM that operates in time over the short clips. Finally, we apply HMR's iterative pose refinement on the output feature vectors from the LSTM for each frame independently. The result is a pose estimate for each frame in the sequence. At training time, we use the simplified version of the HMR loss function that was proposed for training from Kinetics pseudo ground truth [6]. That is, we train directly for Procrustes-aligned 3D keypoint location error (rather than SMPL joint angles/absolute 3D keypoint positions), and 2D reprojection error of the 3D pose.

**Providing motion inputs.** Given a video-based architecture, we next add motion information. Our first strategy is to use optical flow. Optical flow is already known to transfer well across domains, because it relies more on similarities between frames than on recognizing specific patterns [15, 17, 29, 45, 65, 66]. Furthermore, optical flow can have strong cues for depth: for example, if one end of a rigid body is stationary, but the other end is moving toward the first, then this indicates an out-of-plane rotation. We implement this as a simple preprocessing step. That is, we use an off-the-shelf optical flow algorithm FlowNet [15] as a frozen module, which produces an estimate of optical flow at the full resolution of the input sequence.

One disadvantage of optical flow is that it can become difficult to distinguish body parts from background. Especially for frames with little motion, the movement of individual limbs will be simply blobs of smooth motion, much like blobs of background. We hypothesize that this is mostly a problem of 2D part detection, and we note that 2D keypoint detection is a well-studied field. 2D keypoint detection is far easier to annotate than 3D, and even when 2D keypoints are not available, some recent works have argued that 2D correspondence and keypoints can even be obtained in a self-supervised manner [30, 95]. 2D keypoints alone do contain some information about 3D pose [44], although follow-up work has suggested that this approach to using 2D keypoints by itself performs poorly for 3D pose estimation on 3DPW [35]. This leads to an interesting research question: if neither flow information nor 2D keypoints are enough to perform 3D pose estimation, then is it sufficient to identify the 2D keypoints, and then use the flow and keypoint motion to estimate 3D structure?

To answer this question, we provide 2D keypoints as another input to Motion HMR. At training time, these are obtained automatically from the known synthetic pose; at test time, these are the detections from an automatic 2D keypoint detector (for reproducibility, we use the automatic 2D keypoints provided with the 3DPW dataset). After computing optical flow, we concatenate an additional set of 12 channels to the input image which are keypoint heatmaps. That is, each channel contains zeros everywhere except near the keypoint associated with the channel; they are 1 at the keypoint location and fall off with a Gaussian distribution with a standard deviation of 10 pixels. For more details on the architecture and training, see the supplementary material.

## 4 Results

We apply our trained models to the 3D Poses in the Wild dataset. This dataset is challenging because it is shot in real-world environments using handheld cameras, rather than the motion-capture rigs of prior 3D pose estimation work. There is non-trivial camera motion, strong lighting variations (indoor and outdoor scenes), and substantial clutter, including objects moving in the background and occlusions by both objects and humans.

Following prior work [6], we evaluate only on sequences in the test set, and among these, only on frames where at least 7 keypoints are visible (although all frames are visible to the algorithm). We pass 31-frame clips to the algorithm, the same as at training time, and evaluate the predicted poses using the 14 joints that are common to the SMPL and COCO models. We use the standard performance metric PA-MPJPE, which uses the procrustes algorithm to align the poses in 3D before computing squared error, and we average across each individual person before finally averaging across the entire dataset.

As a baseline, we also ran Domain Adversarial Neural Networks (DANN) [18], a mainstay of domain adaptation, which uses an adversarial network trained to distinguish between the representations of real and synthetic images. The trunk of the network is then trained with the negative of this discriminator loss, resulting in representations that are indistinguishable across domains. While straightforward to implement, DANN is challenging to tune: synthetic images and real ones are mapped to overlapping distributions, there is no way to guarantee that the mapping preserves semantics. We apply DANN to the per-frame representations directly before the LSTM.

Table 1 shows our results, comparing training with and without motion information as input to the network. RGB alone performs poorly: a network trained only on short RGB video clips fares worse than one trained on flow, despite the relatively uninformative flow images. This relationship holds even when the RGB-only model is trained with DANN. One possible explanation is that neural networks tend to rely heavily on texture cues [19]. Synthetic textures are not similar to real ones; therefore the network can localize a synthetic person by distinguishing between sim and real textures.

Table 1: Results for sim2real transfer for the 3DPW dataset. Lower is better. Methods above the double line do not train the 3D pose estimator on any real RGB data. We see a substantial boost for using 2D keypoints (also used in Martinez et al.), and non-trivial improvements for using flow rather than RGB. For DANN, we ran 5 seeds with varying weight for the DANN loss, and report the best seed with the mean in parentheses.

| | Algorithm | PA-MPJPE |
|---|---|---|
| Training:<br>3D poses only<br>for synthetic data | Martinez et al. [44] (from [35]) | 157.0 |
| | RGB Only | 105.6 |
| | RGB + DANN [18] | 103.0 (107.5) |
| | Flow Only (proposed) | 100.1 |
| | RGB + Keypoints (proposed) | 82.4 |
| | Keypoints Only (proposed) | 77.6 |
| | Flow + Keypoints (proposed) | 74.7 |
| Training:<br>3D poses<br>on real data | HMR [34] (from [6]) | 77.2 |
| | Temporal HMR [35] | 80.1 |
| | Temporal HMR + InstaVariety [35] | 72.4 |
| | HMR + Kinetics [6] | 72.2 |

At test time this is impossible, and failures to identify parts early on can amplify at later layers which do detailed depth estimation.

Keypoints, on the other hand, perform surprisingly well, validating the intuitions from psychology that 2D motion encapsulates substantial information about 3D activities [32]. It is interesting to compare our keypoints-only results to Martinez *et al.* [44], which is a comparable algorithm in that it only uses 2D keypoints. There are a number of factors which may explain the relatively poor results of [44]. Primarily, [44] is trained on single frames, whereas we use sequences. Furthermore, Martinez *et al.* gives relatively little attention to occlusions, resulting in a domain gap relative to 3DPW.

Another interesting result is that the network which incorporates RGB on top of keypoints actually performs worse than one that uses only keypoints, further emphasizing the size of the domain gap. It's likely the simple presence of synthetic textures causes the network to rely on them, and ignore 2D motion cues that are more reliable out-of-domain, but harder to learn. On the other hand, adding flow to keypoints yields substantial improvements. This suggests that optical flow contains information about motion and silhouettes that pure 2D keypoints do not, and furthermore, that the optical flow estimated on synthetic and real images are a reasonably good match. We conjecture that the flow features lose low-level texture information that neural networks can easily overfit to, replacing it with simple piecewise-smooth regions that capture only shape and motion.

Our final result, of 74.7, is comparable to state-of-the-art works that use similar training pipelines. We outperform HMR [34], which trains on real-world motion capture datasets as well as real-world 2D images as a regularizer (ensuring that 3D poses are consistent with 2D annotations). In contrast, our network is trained *only* on annotated synthetic images from SURREAL. Even more interesting is extensions to HMR that use temporal sequences [35]. While [35] reports pose sequences that are substantially more coherent, they find that adding temporal information *harms* the method's absolute pose accuracy. While counter-intuitive, we hypothesize that this is due to another form of domain gap: specifically, the 3D datasets that this algorithm was trained on all contain static backgrounds. Thus, in the training set, motion is a very strong cue for the person's location. At test time, however, 3DPW contains substantial background motion, which can confuse the algorithm. Our synthetic pipeline, on the other hand, allows us to provide realistic background motion. Overall, the only algorithms which currently outperform us are trained on large, weakly-labeled video datasets [6, 35]. This sort of semi-supervised learning on real videos is an interesting avenue for future research, since it would allow us to add real videos to our synthetic training set without any manual annotation, and therefore potentially boost the performance of our algorithm even further.

Figure 5 shows qualitative results of the performance of our algorithm on 3DPW scenes. Our algorithm is often robust to both unusual poses and to occlusion, even when the occluders are other people (bottom left). For a qualitative comparison to an RGB baseline, as well as outputs on selected whole videos, see the supplementary material.

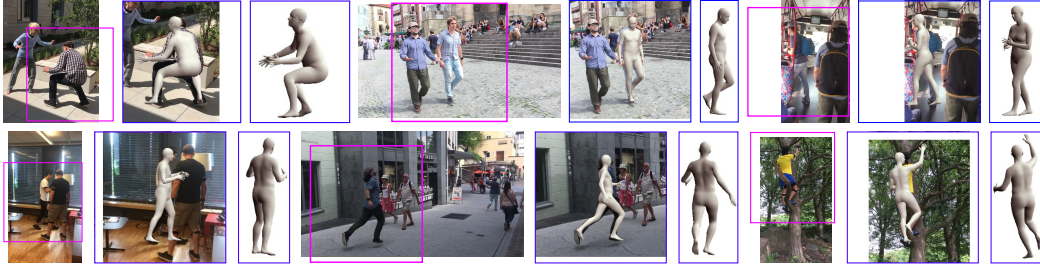

Figure 5: Qualitative results. The input is shown with a magenta detection box identifying the person at its center. Blue boxes show the output of our Motion HMR model. We show both the image-aligned estimated mesh, and the same mesh rotated by 60 degrees. Note that our algorithm is robust to both unusual poses (top left; bottom right) and occlusions (top right; bottom left).

Table 2: Effects of the stages of our dataset generation pipeline.

| Dataset construction approach | PA-MPJPE |
|---|---|
| Full Model | 74.7 |
| No occlusions | 77.2 |
| No background tracking, no occlusions | 80.3 |
| Static background, no occlusions | 88.9 |

Table 3: Effects of changing the clip length used for training and validation.

| Clip length | PA-MPJPE |
|---|---|
| 8 | 79.0 |
| 16 | 77.8 |
| 31 | 74.7 |
| 56 | 76.2 |

## 4.1 Ablations

The ablation results for the dataset preprocessing steps are given in Table 2. In all cases, we train a model from scratch using both optical flow and 2D keypoints as input. We begin with a model that is composited as SURREAL was: we select a single Kinetics frame as background, and composite all of the SURREAL images from a sequence onto it. Replacing static backgrounds with moving ones gives a substantial boost, confirming that networks which segment a moving person from a static background may fail to generalize to dynamic backgrounds. Tracking the background also helps, suggesting that camera motion is a non-trivial artifact in 3DPW. Finally, the boost from using occlusions validates that superpixels are a good approximation to the occlusions seen in real videos.

Finally, we consider the importance of long-term versus short-term motion for our network by varying the length of the clips that were fed into the network and retraining from scratch. Table 3 shows the results. We can see that performance improves until 31 frames, corresponding to roughly 1 second of video. This isn't surprising because 3DPW contains clips where people are occasionally standing still. However, we don't see any improvement moving to 2 seconds of video. One possible explanation is that errors are accumulating in the LSTM as the sequence length increases, indicating a potential area for future research in architectures. Furthermore, with a batch size of 2, we could only fit 2 clips in GPU memory simultaneously, which reduces the stability of the Batch Norm required by HMR (31 frames uses batch size 3; 8 and 16 used batch size 6).

## 5 Conclusions

Our results show that motion information can help neural networks learn 3D human pose estimation from synthetic images. Human pose estimation is challenging because humans are articulated and deformable, with wide appearance variation, yet they are far from the only thing in the visual world like this. Our results may have wide-ranging applications in, for example, robotics, where both camera motion and object motion (via manipulation) can provide strong cues for object pose. While it is somewhat disappointing that neural networks overfit to the RGB appearance of synthetic images, and therefore our final model loses out on cues like shading, it is possible that the advantages of RGB might be recovered through self-supervised learning. That is, we can estimate poses in video using sim2real, potentially fix errors using e.g. bundle adjustment [6], and then train a single-frame RGB model on the result. Overall, we believe motion information, and the sim2real transfer that it enables, may become an essential component of pose estimation systems whenever video is available.

**Acknowledgements:** We thank Konstantinos Bousmalis, João Carreira, Ankush Gupta, Mateusz Malinowski, Relja Arandjelović, Jean-Baptiste Alayrac, Viorica Pătrăucean, Jacob Walker, Yuxiang Zhou, and Anurag Arnab for helpful discussions.

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
