[Supplementary Material]

# Sim2real transfer learning for 3D human pose estimation: motion to the rescue: supplementary file

**Carl Doersch**[*]          **Andrew Zisserman**[*†]

[*] Deepmind, London          [†] VGG, Department of Engineering Science, University of Oxford

## 1   Qualitative video outputs

The qualitative_results folder in this supplementary file contains the output of our algorithm applied to selected 3DPW videos. We show the algorithm's input—optical flow and 2D keypoint maps—as well as the final 3D pose output rendered from multiple views. Note that the motion of the 2D keypoints is often enough to obtain a rough guess at the activity and pose, but without any kind of silhouette it's often difficult to guess the fine joint angles. Flow, on the other hand, provides a reliable cue for the person's silhouette, but is often missing parts of the body which aren't moving.

## 2   Qualitative comparison to RGB-only baseline

For the qualitative results shown in the paper, we here show a comparison to an RGB-only model trained on synthetic humans. Our algorithm is often robust to both unusual poses and to occlusion, even when the occluders are other people (bottom left). On the other hand, the baseline fails badly even for relatively simple poses, again confirming our suspicion that transfer from RGB is so poor that the algorithm cannot even identify the 2D locations of joints. Interestingly, there are similarities between the wrongly-estimated pose (note the elbow angles), suggesting some similarity in the way that real human textures are misinterpreted by the network.

## 3   Implementation details for dataset generation

We randomly crop the Kinetics videos to be $240 \times 320$, to match the SURREAL resolution, potentially resizing so that one dimension exactly matches the SURREAL resolution, and then randomly extract a 31-frame clip from both Kinetics and SURREAL. For every frame, we compute optical flow of the Kinetics data using TVL1 [3], computed offline. Then we compute the median x and y flow for each frame. Finally, we translate the person according to the median flow at each frame, using the center frame of the clip as a reference.

We generate SLIC superpixels directly in the 4-D video tensor. Recall that SLIC is a clustering in RGB/XYZ space. This is sub-optimal under large camera motions, as the clusters are artificially encouraged to remain in the same place relative to the image frame. Therefore, we modify the standard SLIC algorithm by first integrating the median flow vector that we used initially in order to get an estimate of global camera motion throughout the video, and then subtracting the integrated value from the X/Y coordinates used by SLIC. The result is SLIC superpixels of color blobs in the video that roughly follow camera motion, which are inexpensive to compute. Given these superpixels, we choose a random superpixel, and mask out any part of the synthetic human covered by it. Note that, if the superpixel does not occlude the person, or if it occludes so much of the person that there are an average of less than 7 keypoints (out of the 14 standard COCO joints) per frame remaining, then we simply discard the superpixel and do no masking. This means that only roughly 30% of videos have occlusions generated in this way. When computing the superpixel, we use between 10 and 30 superpixels per image. We implement SLIC using an off-the-shelf implementation from skimage, where we add extra channels containing the (x,y) pixel coordinates which have been modified to

Figure 1: Qualitative results. The input is shown with a magenta detection box identifying the person at its center. Blue boxes show our method with 2D keypoints and optical flow; red boxes show the results of the baseline trained only on RGB. We show both the image-aligned estimated mesh, and the same mesh rotated by 60 degrees.

account for camera motion, and then set a very low compactness for the SLIC computation (0.01). The concatenated (x,y) coordinates are multiplied by a random scalar between 4e-4 and 6e-4. To compute the final composite, we use hard binary masks derived from the ground-truth segmentation in SURREAL.

Bounding boxes are cropped at a $224 \times 224$ resolution, following the procedure from HMR which uses the keypoints to ensure that the person is centered and roughly 150 pixels tall. The 'total occlusions' are up to 15 frames long in our training set. Such 'total occlusions' are also used at evaluation time for any frame where fewer than 7 keypoints are visible; note that such frames are not counted in the evaluation.

## 4  Implementation details for network training

The 12 keypoints we use are the ankles, knees, hips, shoulders, elbows, and wrists, which are standard in most pose datasets. We compute FlowNet-based optical flow at the level of bounding boxes to avoid wasting computation on the background. To compute the flow, then, we need two frames for each bounding box. We obtain these by extending the bounding box at each frame into the future to create a set of 'paired boxes' for optical flow. One drawback of this approach is that there is no 'paired box' for the final frame of a clip. To deal with this problem, we predict *two* poses from each pair of frames: one for the present frame and one for the frame in the future. In this way, a 31-frame clip becomes 30 bounding box pairs, and then 60 total predictions. While we compute a loss for all 60 predictions at training time, at test time, we throw out all 'future' predictions except the final one, in order to get 31 predictions.

Our LSTM contains 1024 hidden units, and we add another layer of 1024 units on top of this for each of the prediction heads for the 2 predictions (current and future) that must be made for each frame.

Because keypoint detectors do not reliably detect all keypoints, we find that domain randomization [1, 2] is useful for generating hidden keypoints at training time. That is, we randomly hide keypoints with some probability. All keypoints outside the frame of the kinetics video are considered hidden. All keypoints occluded by the SLIC mask are also considered hidden, as well as keypoints on frames that are totally occluded. Finally, any keypoints that are at least 20cm behind the depth map of the mesh in 3D are considered hidden, because they likely indicate that a limb is occluded behind the body. We then unhide all leg keypoints with a 50% probability, as we find that the 3DPW keypoint detector often produces estimates for leg keypoints even when they are occluded. We then randomly

unhide hidden keypoints and hide un-hidden keypoints with a 5% probability. We initialize HMR using a standard ImageNet-pretrained ResNet-50 model for all experiments (as was done in the original HMR); when using optical flow, we expand 2 channels to 3 by adding an extra channel which is the flow magnitude, and pass this to the ResNet. Optical flow estimates are divided by 20 to put them in the same range as RGB pixels (roughly -1 to 1). When training with keypoint channels, these are concatenated alongside the other inputs, and extra weights in conv1 are initialized randomly.

Our implementation of DANN uses a 2-layer (1024 hidden units) adversary on the ConvNet output (right before the LSTM). We performed a hyperparameter sweep with 5 seeds, using a DANN loss weight between 0.2 and 5 (we found values outside this range performed worse) and report the best, with the mean in parentheses. This is the only experiment for which we performed any hyperparemter sweep.