[Reviews · NeurIPS 2019]

Reviewer 1



Summary: The authors defend in this paper that motion is an effective way to bridge the gap between real and synthetic data. To that end, they use optical flow together with 2D keypoints as input for a network that estimates 3D pose from outdoor images. The system itself is relatively straightforward: they extend HMR with an LSTM and make it process batches of 31 frames containing 2D keypoints and flow. The data is taken from an existing dataset (SURREAL), but is extended to include realistic background motion and occlusions. The system is evaluated on the 3DPW dataset, where they outperform other systems trained without real RGB data, and perform similarly to the state of the art using real RGB. The authors also provide extensive ablation experiments and some qualitative results. Positive: The paper is well written, reproducible, and the system it describes is simple. It contains a very complete and up to date description of the related work. Although the idea of using motion to bridge the gap between synthetic and real data is not new, implementing it in a deep net is, and achieving results that are pretty comparable with state of the art from outdoor images is new as well. Apart from comparing with a number of relevant systems, the experiments provide an ablative evaluation that informs the reader about which parts of the algorithm are most important. Negative: As previously mentioned, the idea is not extremely novel. But its application to a working and well performing deep network system is good, so this is a minor negative factor. Apart from there are just minor things: - Which flow method is used? Flownet is mentioned in the paper (line 62) but TVL1 in supp. matt. (line 20) - In the paragraph 78-85 lines, seems like 59 and 69 references are swapped

Reviewer 2



Originality: the approach is a combination of existing approaches and ideas with a slight modification to enable temporal processing. Through citation of a large corpus of work the authors make that very clear. quality: the paper is of high quality and thoughfully written. I appreciated the detailed evaluation with a thorough ablation study for the model as well as the dataset. clarity: The paper is well and clearly written. The motivation is clearly stated, and the approach clearly described. significance: showing results of a model trained on simulated data that performs on par with models trained on real data is an important contribution. Various question: 1) Fig 5 is hard to understand and follow. Some images do show detections some dont? Why are some blue boxes empty? 2) The authors mention and compare against the Temporal HMR [35] model - it would be important to clarify the difference to the Motion HMR model. They sound very similar. 3) The paper title is very general about 3D pose estimation. While I agree that insights should be useful for training more general object tracking systems beyond humans, the paper does only show results for human 3D pose tracking. I would strongly encourage a modification to the title to reflect the fact that the paper only deals with human 3D pose estimation.

Reviewer 3



Positives The paper is well-written and includes a through literature review. The following paper is also very relevant to the submission: Shrivastava, Ashish, et al. "Learning from simulated and unsupervised images through adversarial training." Proceedings of the IEEE conference on computer vision and pattern recognition. 2017. Novelty of the method over [44] is not major. Still, I believe no one has shown that computing flow on simulated data and using it for training improves over RGB only (although the improvement is quite marginal). Simulation pipeline proposed in the paper seems to be quite useful. It improves over the previous approaches that used simpler compositing operations. Negatives My main concern is about the generality of the method: 1. It only applies to video data which is a major limitation. 2. Using only simulated data (flow only), accuracy is still very far from the state of the art. The method requires additional information from RGB images in the form of 2D keypoints to improve the result (In fact 2D keypoints only has almost the same performance). This network is trained using supervised data. This is not coherent with the claim of the paper that the method completely relies on simulated human data. My second concern is whether the method is a viable solution for sim2real problem. Flow only is only marginally better than RGB only (105.6 vs. 100.1) which introduces a doubt about motion computed using simulated data being a viable solution for sim2real problem. Overall, the paper has good points but I believe the negatives mentioned above weighs more. I encourage the authors to address the concerns above. ------ Revision after rebuttal: Going over my review, together with other reviews and rebuttal, I think the paper deserves a marginally above average rating which I will revise. Nevertheless, I stand by my original review points, particularly about the generality of the method: 1. To clarify, one of the limitations I stated is requiring the motion of the humans and/or camera, not only application to video. Compared to many of the methods that the paper is compared against (e.g. in Table 1, Martinez et al., DANN or HMR), this is a limitation. There is no question about working on video data being an important problem. 2. I agree that the paper is not trying to hide it is using real data supervision for keypoint detection. However, this fact still weakens the main claim of the paper which is “… motion can be a simple way to bridge a sim2real gap”. I appreciate that the paper presents an optimized pipeline which combines real and synthetic data to obtain a good 3D human pose estimation result, however in my opinion “… motion can be a simple way to bridge a sim2real gap” is a strong claim that is not strongly supported by the experiments presented in the paper. In addition, as also noted in the rebuttal, I would recommend the paper rephrases some of the claims about simulated data such as in abstract: “… on par with state-of-the-art methods trained on real 3D sequences despite training only on synthetic humans from the standard SURREAL dataset”.

[Author Response · NeurIPS 2019]

We thank the reviewers for their constructive feedback. Overall, the reviewers (especially R1 and R2) appreciated our algorithm's simplicity, its effectiveness on challenging real-world 3D human pose estimation, and our thorough experiments. R1 and R3 do point out that the technical novelty of the method is minor: indeed, most of the techniques we use to extract motion and build our dataset are familiar. R1 rightly points out that these techniques are not really the goal of the paper: instead, our goal was to investigate which pieces of the synthetic training pipeline are most important for success, and we found surprising results with a relatively simple, understandable methodology.

R3, on the other hand, mentions problems with "generality" and "viability" as significant weaknesses. On generality, R3's first concern is that the learning method is limited to video data. This concern is puzzling to us. Many algorithms are developed solely for video (e.g. self-supervised methods like 'Shuffle and Learn[1]'), and for many domains of interest in sim2real (e.g. robotics), video is readily available. But more fundamentally, our algorithm allows pseudo-labeling of individual frames, which means that a single-frame model can be learned from unlabeled data. This is indeed future research, but it's premature to say that our approach doesn't apply at all to single-frame situations.

R3's second concern is that our 2D keypoint detector has been trained from annotations on real images. We are not trying to hide this fact. We are asserting that we train the full 3D pose estimator only on synthetic humans; that is, we use no ground-truth depth for real images. We are not wishing to claim (as stated by R3) that "the method completely relies on simulated human data." We have identified a few places where our original prose may have been unclear about this, and we will rewrite them. Importantly, while our ultimate goal is to remove manual annotation from the pipeline completely, the field is currently far from this goal. Therefore, progress will require gradually reducing reliance on labeled real data, starting with the most difficult parts of the pipeline like depth. Our work is a core step on this path.

Finally, in evaluating our results, R3 almost completely ignores our contributions showing *how* to get good performance when using 2D keypoints as input, seemingly treating data generation and optical flow as the only contributions of our method. Apparently this is because the "Novelty of the method over [44] is not major", where [44] also uses keypoints. In response, we point out that our algorithm better than halves the error achieved by [44] on 3DPW. The main difference is that [44] does not use *motion* information, the importance of which is the core thesis of our work. Finally, even if focusing on flow, an improvement of 5.5 on 3DPW is hardly marginal: [6] for example, shows an improvement of 5.0 with their weak labeling method. It only appears marginal relative to the enormous loss in performance that comes from naïve use of synthetic data, and the performance recovery from keypoints.

**Detailed responses:** R1: FlowNet vs TVL1: We used two different algorithms only for implementation convenience. We had TVL1 features pre-computed for Kinetics, and so it was straightforward to use these to estimate camera motion. However, we didn't have TVL1 flow precomputed for the composited videos, so we did this on-the-fly, and FlowNet was easiest to implement in our Tensorflow scripts. If required, we can re-run using FlowNet for everything; we expect results to be very similar.

R1: references 59 and 69 are swapped: This is indeed an error. Thanks for pointing it out; we'll fix it in the final manuscript.

R2: Fig. 5 is hard to understand: For each test image, we show 3 displays: the original image with the detection (which all have bounding boxes), the extracted box with the inferred mesh, and the rotated mesh. In retrospect, we agree that this presentation makes it difficult to group the images. We'll improve it in the final version. However, there should also be no empty blue boxes; this suggests an issue with the reader. We can debug if you tell us which one you used.

R2: Differences between Temporal HMR [35] and Motion HMR: Architecturally, the methods are intentionally similar for comparability. However, one difference is that [35] uses a 3D convnet, while we use an LSTM to aggregate in time: this is because we expect optical flow to capture short-range information, and care more about propagating pose information long-term across frames without motion. However, between the methods, the more fundamental difference is the training data (synthetic vs. real), and we expect these differences to be more important for the overall behavior of the model.

R2: Title is general: We wrote the title thinking that our method can be readily applied to other types of 3D pose estimation whenever synthetic data is available, and would therefore be of interest to all 3D pose estimation researchers. However, we are happy to update the title as requested.

R3: Reference to Shrivastava, Ashish, et al.: Thanks for this reference; we weren't aware of this paper at submission time, but will be happy to add it.

# References

[1] I. Misra, C. L. Zitnick, and M. Hebert. Shuffle and learn: unsupervised learning using temporal order verification. In *ECCV*, 2016.


[Meta-Review · NeurIPS 2019]

After reviewer discussion and rebuttal this paper received three acceptance recommendations. R1 and R2 are more positive about the paper and acknoweldge the contribution. R3 points out that the impact of using just flow and no person and camera motion is limited. Please consider the post-rebuttal portion of the review to include in a final revision. The method, approach and quality of the paper are high as acknowledged by all reviewers. The only disagreement is on the significance part. The problem of learning from simulated data is relevant and applies to the problem studied. This submission presents a system that makes a step forward in this direction, although from an empirical perspective it may be smaller than anticipated. The empirical results appear conclusive though and all reviewers appreciate the ablation studies that are likely to lead to insights for other works. In summary the reviewers are all on the positive side, although one reviewer being more borderline-positive. The positive aspects of this submission outweigh the negative ones on significance (and slightly novelty), the paper is of high quality and should be accepted.